# OPTIMIZING LARGE-SCALE HYPERPARAMETERS VIA AUTOMATED LEARNING ALGORITHM

## ABSTRACT

Modern machine learning algorithms usually involve tuning multiple (from one to thousands) hyperparameters which play a pivotal role in terms of model generalizability. Black-box optimization and gradient-based algorithms are two dominant approaches to hyperparameter optimization while they have totally distinct advantages. How to design a new hyperparameter optimization technique inheriting all benefits from both approaches is still an open problem. To address this challenging problem, in this paper, we propose a new hyperparameter optimization method with zeroth-order hyper-gradients (HOZOG). Specifically, we first exactly formulate hyperparameter optimization as an $\mathcal{A}$-based constrained optimization problem, where $\mathcal{A}$ is a black-box optimization algorithm (such as deep neural network). Then, we use the average zeroth-order hyper-gradients to update hyperparameters. We provide the feasibility analysis of using HOZOG to achieve hyperparameter optimization. Finally, the experimental results on three representative hyperparameter (the size is from 1 to 1250) optimization tasks demonstrate the benefits of HOZOG in terms of *simplicity, scalability, flexibility, effectiveness and efficiency* compared with the state-of-the-art hyperparameter optimization methods.

## 1 INTRODUCTION

Modern machine learning algorithms usually involve tuning multiple hyperparameters whose size could be from one to thousands. For example, support vector machines (Vapnik, 2013) have the regularization parameter and kernel hyperparameter, deep neural networks (Krizhevsky et al., 2012) have the optimization hyperparameters (e.g., learning rate schedules and momentum) and regularization hyperparameters (e.g., weight decay and dropout rates). The performance of the most prominent algorithms strongly depends on the appropriate setting of these hyperparameters.

Traditional hyperparameter tuning is treated as a bi-level optimization problem as follows.

$$\min_{\lambda \in \mathbb{R}^p} f(\lambda) = E(w(\lambda), \lambda), \quad s.t. \ w(\lambda) \in \arg\min_{w \in \mathbb{R}^d} L(w, \lambda) \tag{1}$$

where $w \in \mathbb{R}^d$ are the model parameters, $\lambda \in \mathbb{R}^p$ are the hyperparameters, the outer objective $E$ [1] represents a proxy of the generalization error w.r.t. the hyperparameters, the inner objective $L$ represents traditional learning problems (such as regularized empirical risk minimization problems), and $w(\lambda)$ are the optimal model parameters of the inner objective $L$ for the fixed hyperparameters $\lambda$. Note that the size of hyperparameters is normally much smaller than the one of model parameters (*i.e.*, $p \ll d$). Choosing appropriate values of hyperparameters is extremely computationally challenging due to the nested structure involved in the optimization problem. However, at the same time both researchers and practitioners desire the hyperparameter optimization methods as *effective*, *efficient*, *scalable*, *simple* and *flexible*[2] as possible.

Classic techniques such as grid search (Gu & Ling, 2015) and random search (Bergstra & Bengio, 2012) have a very restricted application in modern hyperparameter optimization tasks, because they only can manage a very small number of hyperparameters and cannot guarantee to converge to

---

[1]The choice of objective function $E$ depends on the specified tasks. For example, accuracy, AUC or F1 can be used for binary classification problem. Square error loss or absolute error loss can be used as the objective of $E$ for regression problems on validation samples.

[2]"effective": good generalization performance. "efficient": running fast. "scalable": scalable in terms of the sizes of hyperparameters and model parameters. "simple": easy to be implemented. "flexible": flexible to various learning algorithms.

Table 1: Representative black-box optimization and gradient-based hyperparameter optimization algorithms. ("BB" and "G" are the abbreviations of black-box and gradient respectively, and "♣" denotes that the property holds for a small number of hyperparmaters or medium-sized training set. "Scalable-H" and "Scalable-P" denotes scalability in terms of hyperparameters and model parameters respectively.)

| Algorithm | Type | Properties | | | | | |
|---|---|---|---|---|---|---|---|
| | | Effective | Efficient | Scalable-H | Simple | Flexible | Scalable-P |
| GPBO (Snoek et al., 2012) | BB | ♣ | ♣ | ✗ | ✓ | ✓ | ✓ |
| BOHB (Falkner et al., 2018) | BB | ♣ | ♣ | ✗ | ✓ | ✓ | ✓ |
| HOAG (Pedregosa, 2016) | G | ✓ | ✓ | ✓ | ✗ | ✗ | ✗ |
| RMD (Maclaurin et al., 2015) | G | ✓ | ✓ | ✓ | ✗ | ✗ | ✗ |
| RFHO (Franceschi et al., 2017; 2018) | G | ✓ | ✓ | ✓ | ✗ | ✗ | ✗ |
| HOZOG | BB+G | ✓ | ✓ | ✓ | ✓ | ✓ | ✓ |

local/global minima. For modern hyperparameter tuning tasks, black-box optimization (Snoek et al., 2012; Falkner et al., 2018) and gradient-based algorithms (Maclaurin et al., 2015; Franceschi et al., 2018; 2017) are currently the dominant approaches due to the advantages in terms of *effectiveness, efficiency, scalability, simplicity and flexibility* which are abbreviated as E2S2F in this paper. We provide a brief review of representative black-box optimization and gradient-based hyperparameter optimization algorithms in §2.1, and a detailed comparison of them in terms of the above properties in Table 1.

Table 1 clearly shows that black-box optimization and gradient-based approaches have totally distinct advantages, *i.e.*, black-box optimization approach is simple, flexible and salable in term of model parameters, while gradient-based approach is effective, efficient and scalable in term of hyperparmeters. Each property of E2S2F is an important criterion to a successful hyperparameter optimization method. To the best of our knowledge, there is still no algorithm satisfying all the five properties simultaneously. Designing a hyperparameter optimization method having the benefits of both approaches is still an open problem.

To address this challenging problem, in this paper, we propose a new hyperparameter optimization method with zeroth-order hyper-gradients (HOZOG). Specifically, we first exactly formulate hyperparameter optimization as an $\mathcal{A}$-based constrained optimization problem, where $\mathcal{A}$ is a black-box optimization algorithm (such as the deep neural network). Then, we use the average zeroth-order hyper-gradients to update hyperparameters. We provide the feasibility analysis of using HOZOG to achieve hyperparameter optimization. Finally, the experimental results of various hyperparameter (the size is from 1 to 1250) optimization problems demonstrate the benefits of HOZOG in terms of E2S2F compared with the state-of-the-art hyperparameter optimization methods.

## 2 HYPERPARAMETER OPTIMIZATION BASED ON ZEROTH-ORDER HYPER-GRADIENTS

In this section, we first give a brief review of black-box optimization and gradient-based algorithms, and then provide our HOZOG algorithm. Finally, we provide the feasibility analysis of HOZOG.

### 2.1 BRIEF REVIEW OF BLACK-BOX OPTIMIZATION AND GRADIENT-BASED ALGORITHMS

**Black-box optimization algorithms:** Black-box optimization algorithms view the bilevel optimization problem $f$ as a black-box function. Existing black-box optimization methods (Snoek et al., 2012; Falkner et al., 2018) mainly employ Bayesian optimization (Brochu et al., 2010) to solve (1). Black-box optimization approach has good simplicity and flexibility. However, a lot of references have pointed out that it can only handle hyperparmeters from a few to several dozens (Falkner et al., 2018) while the number of hyperparmeters in real hyperparameter optimization problems would range from hundreds to thousands. Thus, black-box optimization approach has weak scalability in term of the size of of hyperparmeters.

**Gradient-based algorithms:** The existing gradient-based algorithms can be divided into two parts (*i.e.*, inexact gradients and exact gradients). The approach of inexact gradients first solves the inner problem approximately, and then estimates the gradient of (1) based on the approximate solution by the approach of implicit differentiation (Pedregosa, 2016). Because the implicit differentiation involves Hessian matrices of sizes of $d \times d$ and $d \times p$ where $p \ll d$, they have poor scalability. The

approach of exact gradients[3] treats the inner level problem as a dynamic system, and use chain rule (Rudin et al., 1964) to compute the gradient. Because the chain rule highly depends on specific learning algorithms, this approach has poor flexibility and simplicity. Computing the gradients involves Hessian matrices of sizes of $p \times p$ and $d \times p$. Thus, the approach of exact gradients has better scalability than the approach of inexact gradients because normally we have $p \ll d$.

**† Enlightenment:** As introduced in (Nesterov & Spokoiny, 2017; Gu et al., 2018), zeroth-order gradient (also known as finite difference approximation (Cui et al., 2017)) technique is a black-box optimization method which estimates the gradient only by two function evaluations. Thus, zeroth-order gradient technique belongs both to black-box optimization and gradient-based optimization. We hope that the hyperparameter optimization method bases on zeroth-order hyper-gradients[4] can inherit all benefits as described in Table 1.

## 2.2 HOZOG Algorithm

▶ **Principle:** Instead of directly computing the hyper-gradient as in (Pedregosa, 2016; Maclaurin et al., 2015; Franceschi et al., 2017; 2018), we use two function evaluations (*i.e.*, the zeroth-order hyper-gradient technique (Nesterov & Spokoiny, 2017; Gu et al., 2018)) to estimate the hyper-gradient, and update hyperparameters with hyper-gradients which derives our HOZOG algorithm.

Before presenting HOZOG algorithm in detail, we first clarify what problem we are solving exactly.

▶ **What problem we are solving exactly?** As mentioned in (1), the inner level problem in the traditional hyperparameter tuning is finding the model parameters that minimize the inner objective $L$, (*i.e.*, $w(\lambda) \in \arg\min_{w \in \mathbb{R}^d} L(w, \lambda)$). However, in the real-world hyperparameter tuning problems, we are usually trying to find an approximate minimum solution of $L$ by an optimization algorithm if the inner level problem $L$ in convex. If the inner level problem $L$ in non-convex, we usually try to find an approximate local solution or a stationary point. Thus, we replace the inner level problem by $w(\lambda) = \mathcal{A}(\lambda)$ where $\mathcal{A}$ is an optimization algorithm which approximately solves the inner objective $L$. Further, we replace the bi-level optimization problem (1) by the following $\mathcal{A}$-based constrained optimization problem (2).

$$\min_{\lambda \in \mathbb{R}^p} f(\lambda) = E(w(\lambda), \lambda), \quad s.t. \ w(\lambda) = \mathcal{A}(\lambda) \tag{2}$$

where $w(\lambda)$ are the values returned by the optimization algorithm $\mathcal{A}$.

• *Hyperparameters*: Hyperparameters can be divided into two types, *i.e.*, problem-based hyperparameters and algorithm-based hyperparameters.

1. Problem-based hyperparameters: The problem-based hyperparameters are the hyperparameters involved in *learning problems* such as the regularization parameter and the architectural hyperparameters in deep neural networks.
2. Algorithm-based hyperparameters: These are the hyperparameters involved in *optimization algorithms* such as the learning rate, momentum and dropout rates.

The traditional bi-level optimization problem (1) can only formulate the problem-based hyperparameters. However, our $\mathcal{A}$-based constrained optimization problem (2) can formulate both types of hyperparameters.

▶ **Algorithm:** To solve the $\mathcal{A}$-based constrained optimization problem (2), we propose HOZOG algorithm in Algorithm 1, where the "for" loop is referred to as "meta-iteration". We describe the two key operations of Algorithm 1 (*i.e.*, estimating the function value and average zeroth-order hyper-gradient) in detail as follows.

• *Estimating the function value*: We treat the optimization algorithm $\mathcal{A}$ as a black-box oracle. Given hyperparameters $\lambda$, the black-box oracle $\mathcal{A}$ returns model parameters $w(\lambda)$. Based on the pair of $\lambda$ and $w(\lambda)$, the function value can be estimated as $E(w(\lambda), \lambda)$.

• *Computing the average zeroth-order hyper-gradient*: Zeroth-order hyper-gradient can be computed as $\bar{\nabla} f(\lambda) = \frac{p}{\mu} (f(\lambda + \mu u) - f(\lambda)) u$ based on the two function evaluations $f(\lambda + \mu u)$ and $f(\lambda)$,

---

[3]Although the inner-problem is usually solved approximately *e.g.* by taking a finite number of steps of gradient descent, we still call this kind of methods as exact gradients throughout this paper to avoid using too complex terminology.

[4]We call the gradient *w.r.t.* hyperparameter as hyper-gradient in this paper.

---

**Algorithm 1** Hyperparameter optimization method with zeroth-order hyper-gradients (HOZOG)

---

**Input:** Learning rate $\gamma$, approximate parameter $\mu$, size of directions $q$ and black-box inner solver $\mathcal{A}$.
 1: Initialize $\lambda_0 \in \mathbb{R}^p$.
 2: **for** $t = 0, 1, 2, \ldots, T - 1$ **do**
 3:  Generate $u = [u_1, \ldots, u_q]$, where $u_i \sim N(0; I_p)$.
 4:  Compute the average zeroth-order hyper-gradient $\hat{\nabla} f(\lambda_t) = \frac{p}{\mu q} \sum_{i=1}^{q} (f(\lambda_t + \mu u_i) - f(\lambda_t)) u_i$, where $f(\lambda_t)$ is estimated based on the solution returned by the black-box inner solver $\mathcal{A}$.
 5:  Update $\lambda_{t+1} \leftarrow \lambda_t - \gamma \hat{\nabla} f(\lambda_t)$.
 6: **end for**
**Output:** $\lambda_T$.

---

where $u \sim N(0, I_p)$ is a random direction drawn from a uniform distribution over a unit sphere, and $\mu$ is an approximate parameter. $\bar{\nabla} f(\lambda)$ has a large variance due to single direction $u$. To reduce the variance, we use the average zeroth-order hyper-gradient (3) by sampling a set of directions $\{u_i\}_{i=1}^{q}$.

$$\hat{\nabla} f(\lambda) = \frac{p}{\mu q} \sum_{i=1}^{q} (f(\lambda + \mu u_i) - f(\lambda)) u_i \tag{3}$$

Based on the average zeroth-order hyper-gradient $\hat{\nabla} f(\lambda)$, we update the hyperparameters as follows.

$$\lambda \leftarrow \lambda - \gamma \hat{\nabla} f(\lambda) \tag{4}$$

Note that $\hat{\nabla} f(\lambda)$ is a biased approximation to the true gradient $\nabla f(\lambda)$. Its bias can be reduced by decreasing the value of $\mu$. However, in a practical system, $\mu$ could not be too small, because in that case the function difference could be dominated by the system noise (or error) and fails to represent the function differential (Lian et al., 2016).

• *Parallel acceleration.* Because the average zeroth-order hyper-gradient involves $q + 1$ function evaluations as shown in (3), we can use GPU or multiple cores to compute the $q + 1$ function evaluations in parallel to accelerate the computation of average zeroth-order hyper-gradients.

### 2.3 FEASIBILITY ANALYSIS

▶ **Challenge:** In treating the optimization algorithm $\mathcal{A}(\lambda)$ as a black-box oracle that maps $\lambda$ to $w$, the most important problem is whether the mapping function $\mathcal{A}(\lambda)$ is *continuous* which is the basis of using the zeroth-order hyper-gradient technique to optimize (2).

• **Continuity:** Before discussing the continuity of the $\mathcal{A}$-based constrained optimization problem $f(\lambda)$, we first give the definitions of iterative algorithm and continuous function in Definitions 1 and 2 respectively.

**Definition 1** (Iterative algorithm)**.** *Assume the optimization algorithm $\mathcal{A}(\lambda)$ can be formulated as a nested function as $\mathcal{A}(\lambda) = w_T$ and $w_t = \Phi_t(w_{t-1}, \lambda)$ for $t = 1, \ldots, T$, where $T$ is the number of iterations, $w_0$ is an initial solution, and, for every $t \in \{1, \ldots, T\}$, $\Phi_t : (\mathbb{R}^d \times \mathbb{R}^p) \to \mathbb{R}^d$ is a mapping function that represents the operation performed by the t-th step of the optimization algorithm. We call the optimization algorithm $\mathcal{A}(\lambda)$ as an iterative algorithm.*

**Definition 2** (Continuous function)**.** *For all $\lambda \in \mathbb{R}^p$, if the limit of $f(\lambda + \delta)$ as $\delta \in \mathbb{R}^p$ approaches $\mathbf{0}$ exists and is equal to $f(\lambda)$, we call the function $f(\lambda)$ is continuous everywhere.*

Based on Definitions 1 and 2, we give Theorem 1 to show that the $\mathcal{A}$-based constrained optimization problem $f(\lambda)$ is continuous under mild assumptions. The proof is provided in Appendix.

**Theorem 1.** *If the hyperparameters $\lambda$ are continuous and the mapping functions $\Phi_t(w_{t-1}, \lambda)$ (for every $t \in \{1, \ldots, T\}$) are continuous, the mapping function $\mathcal{A}(\lambda)$ is continuous, and the outer objective $E$ is continuous, we have that the $\mathcal{A}$-based constrained optimization problem $f(\lambda)$ is continuous w.r.t. $\lambda$.*

We provide several popular types of optimization algorithms to show that almost existing iterative algorithms are continuous mapping functions which would make $f(\lambda)$ continious.

1. **Gradient descent algorithms**: If $\mathcal{A}$ is a gradient descent algorithm (such as SGD (Ghadimi & Lan, 2013), SVRG (Reddi et al., 2016; Allen-Zhu & Hazan, 2016), SAGA (Defazio et al., 2014),

SPIDER (Fang et al., 2018)), the updating rules can be formulated as $w \leftarrow w - \gamma' v$, where $v$ is a stochastic or deterministic gradient estimated by the current $w$, and $\gamma'$ is the learning rate. To accelerate the training of deep neural networks, multiple adaptive variants of SGD (e.g., Adagrad, RMSProp and Adam (Goodfellow et al., 2016)) have emerged.

2. **Proximal gradient descent algorithms**: If $\mathcal{A}$ is a proximal gradient descent algorithm (Zhao et al., 2014; Xiao & Zhang, 2014; Gu & Huo, 2018), the updating rules should be the form of $w \leftarrow \text{Prox}(w - \gamma' v)$, where Prox is a proximal operator (such as the soft-thresholding operator for Lasso (Tibshirani, 1996)) which is normally continuous (Bredies & Lorenz, 2007; Zou, 2006).

It is easy to verify that the mapping functions $\mathcal{A}(\lambda)$ corresponding to these iterative algorithms are continuous according to Theorem 1.

For a continuous function $f(\lambda)$, there exists a Lipschitz constant $L$ (see Definition 3) which upper bounds $\frac{|f(\lambda_1) - f(\lambda_2)|}{\|\lambda_1 - \lambda_2\|}$, $\forall \lambda_1, \lambda_2 \in \mathbb{R}^p$. Unfortunately, exactly calculating the Lipschitz constant of $f(\lambda)$ is NP-hard problem (Virmaux & Scaman, 2018). We provide an upper bound[5] to the Lipschitz constant of $f(\lambda)$ in Theorem 2.

**Definition 3** (Lipschitz continuous constant). *For a continuous function $f(\lambda)$, there exists a constant $L$ such that, $\forall \lambda_1, \lambda_2 \in \mathbb{R}^p$, we have $\|f(\lambda_1) - f(\lambda_2)\| \leq L\|\lambda_1 - \lambda_2\|$. The smallest $L$ for which the inequality is true is called the Lipschitz constant of $f(\lambda)$.*

**Theorem 2.** *Given the continuous mapping functions $\Phi_t(w_{t-1}, \lambda)$ where $t \in \{1, \ldots, T\}$), $A_t = \frac{\partial \Phi_t(w_{t-1}, \lambda)}{\partial w_{t-1}}$, $B_t = \frac{\partial \Phi_t(w_{t-1}, \lambda)}{\partial \lambda}$. Given the continuous objective function $E(w_T, \lambda)$, $A_{T+1} = \frac{\partial E(w_T, \lambda)}{\partial w_T}$ and $B_{T+1} = \frac{\partial E(w_T, \lambda)}{\partial \lambda}$. Let $L_{A_t} = \sup_{\lambda \in \mathbb{R}^p, w \in \mathbb{R}^d} \|A_{t+1}\|_2$, $L_{B_t} = \sup_{\lambda \in \mathbb{R}^p, w \in \mathbb{R}^d} \|B_t\|_2$. Let $L(f)$ denote the Lipschitz constant of the continuous function $f(\lambda)$, we can upper bound $L(f)$ by $\sum_{t=1}^{T+1} L_{B_t} L_{A_{t+1}} \ldots L_{A_{T+1}}$.*

▶ **Conclusion:** Because the $\mathcal{A}$-based constrained optimization problem $f(\lambda)$ is continuous, we can use the zeroth-order hyper-gradient technique to optimize $f(\lambda)$ (Nesterov & Spokoiny, 2017). Nesterov & Spokoiny (2017) provided the convergence guarantee of zeroth-order hyper-gradient method when $f(\lambda)$ is Lipschitz continuous as defined in Definition 3.

## 3 EXPERIMENTS

We conduct the hyperparameter optimization experiments on three representative learning problems (*i.e.*, $l_2$-regularized logistic regression, deep neural networks (DNN) and data hyper-cleaning), whose sizes of hyperparameters are from 1 to 1250. We also test the parameter sensitivity analysis of HOZOG under different settings of parameters $q$, $\mu$ and $\gamma$, which are included in Appendix due to the page limit. All the experiments are conducted on a Linux system equipped with four NVIDIA Tesla P40 graphic cards.

• **Compared algorithms:** We compare our HOZOG with the representative hyperparameter optimization approaches such as random search (RS) (Bergstra & Bengio, 2012), RFHO with forward (FOR) or reverse (REV) gradients (Franceschi et al., 2017) [6], HOAG (Pedregosa, 2016)[7], GPBO Snoek et al. (2012) [8] and BOHB (Falkner et al., 2018) [9]. Most of them are the representative black-box optimization and gradient-based hyperparameter optimization algorithms as presented in Table 1. We implement our HOZOG in Python[10].

• **Evaluation criteria:** We compare different algorithms with three criteria, *i.e.*, $\|\nabla f(\lambda)\|_2$, suboptimality and test error, where "suboptimality" denotes $f(\lambda) - f(\lambda^\diamond)$ and $f(\lambda^\diamond)$ is the minimum

---

[5]Although the upper bound is related to $T$, our simulation results show that it does not grow exponentially with $T$ because $L_{A_t}$ or $L_{B_t}$ is not larger than one at most times.

[6]The code of RFHO is is available at https://github.com/lucfra/RFHO.

[7]The code of HOAG is available at https://github.com/fabianp/hoag.

[8]The code of GPBO is available at http://github.com/fmfn/BayesianOptimization/.

[9]The code of BOHB is available at https://github.com/automl/HpBandSter. Note that BOHB is an improved version of Hyperband (Li et al., 2017). Thus, we do not compare HOZOG with Hyperband.

[10]We will release the code of HOZOG and the experiments after the paper is accepted.

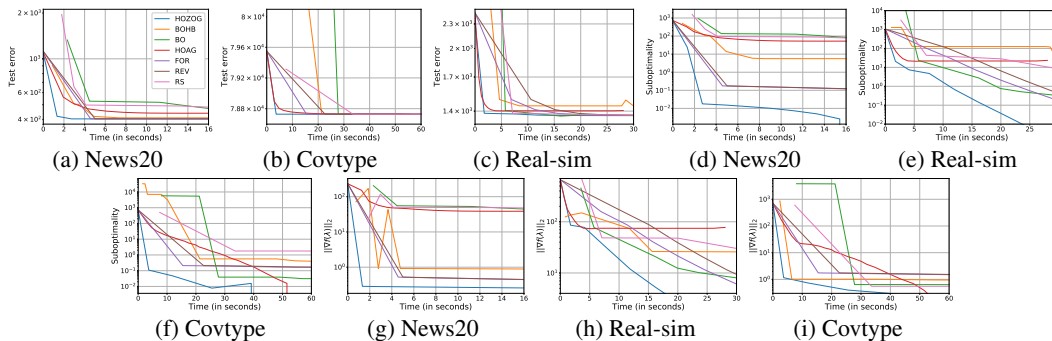

Figure 1: Comparison of different hyperparameter optimization algorithms for $l_2$-regularized logistic regression sharing the same legend. (a)-(c): Test error. (d)-(f): Suboptimality. (g)-(i): $\|\nabla f(\lambda)\|_2$. (Larger figures can be found in the supplement material.)

value of $f(\lambda)$ for all $\lambda$ which have been explored, and test error is the average loss on the testing set. Note the hyper-gradients $\nabla f(\lambda)$ for all method except for FOR and REV are computed by Eq. (3).

• **Datasets:** The datasets used in experiments are News20, Covtype, Real-sim, CIFAR-10 and Mnist datasets from LIBSVM repository, which is available at `https://www.csie.ntu.edu.tw/~cjlin/libsvmtools/datasets/`. Especially, for News20 and Mnist two multi-class datasets, we transform them to binary classification problems by randomly partitioning the data into two groups.

| Experiment | | # HP | Dataset | $q$ | $\mu$ | $\gamma$ |
|---|---|---|---|---|---|---|
| $l_2$-regularized logistic regression | | 1 | News20 | 1 | 0.01 | 0.05 |
| | | | Covtype | 1 | 0.01 | 0.03 |
| | | | Real-sim | 1 | 0.01 | 0.005 |
| Deep Neural Networks | 2-layer CNN | 100 | | 1 | 1 | 1 |
| | VGG-16 | 20 | CIFAR-10 | 3 | 1 | 1 |
| | ResNet-152 | 10 | | 3 | 1 | 1 |
| Data hyper-cleaning | | 500/1250 | Mnist | 5 | 1 | 1 |

Table 2: The parameter settings of HOZOG in the experiments. ("# HP" is the abbreviation of the number of hyperparameters.)

• **Parameters of HOZOG:** The values of parameters $q$, $\mu$ and $\gamma$ in HOZOG are given in Table 2. Especially, $q$ plays an important role to HOZOG because it determines the accuracy and the running time of estimating the gradients. We empirically observe that $q \leq 5$ has a good balance between the two objectives.

## 3.1 $l_2$-REGULARIZED LOGISTIC REGRESSION

**Experimental setup:** We consider to estimate the regularization parameter in the $l_2$-regularized logistic regression model. We split one data set into three subsets (*i.e.*, the train set $\mathcal{D}_{tr}$, validation set $\mathcal{D}_{val}$ and test set $\mathcal{D}_t$) with a ratio of 2:1:1. We use the logistic loss $l(t) = \log(1 + e^{-t})$ as the loss function. The hyperparameter optimization problem for $l_2$-regularized logistic regression is formulated as follows.

$$\underset{\lambda \in [-10,10]}{\arg\min} \sum_{i \in \mathcal{D}_{val}} l(y_i\langle x_i, w(\lambda)\rangle), \qquad s.t. \quad w(\lambda) \in \underset{w \in \mathbb{R}^d}{\arg\min} \sum_{i \in \mathcal{D}_{tr}} l(y_i\langle x_i, w(\lambda)\rangle) + e^\lambda\|w\|^2 \quad (5)$$

The solver used for solving the inner objective is L-BFGS[11] (Liu & Nocedal, 1989) for HOAG and Adam (Kingma & Ba, 2014) for the others.

**Results and discussions:** Figure 1 presents the convergence results of suboptimality, $\|\nabla f(\lambda)\|_2$ and test error *vs.* the running time for different methods. Note that we take same initial values of $\lambda$ and $w$ for all gradient-based methods, while the black-box methods naturally start from different points. Because HOAG works with tolerances and warm start strategy, HOAG has a fast convergence at the early stage but a slow convergence at the late stage as shown in Figures 1g-1i. We observe that HOZOG runs faster than other gradient-based methods. This is because that FOR and REV need much time to compute hyper-gradients. Figures 1g-1i provide $\|\nabla f(\lambda)\|_2$ of different methods as functions of running time. We can see that the black-box methods (*i.e.*, BOHB and GPBO) spend much time on exploring because $\|\nabla f(\lambda)\|_2$ of these methods didn't strictly go down in the early stage. Overall, all the results show that HOZOG has a faster convergence than other methods.

## 3.2 DEEP NEURAL NETWORKS

**Experimental setup:** We validate the advantages of HOZOG on optimizing learning rates of DNN which is much more complicated in both structure and training compared to $l_2$-regularized logistic regression.

---

[11]The implementation is available at `https://github.com/fabianp/hoag`.

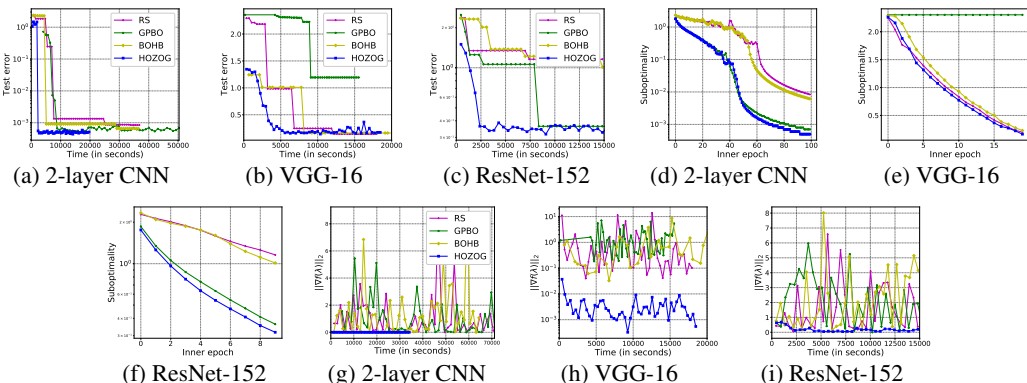

(a) 2-layer CNN    (b) VGG-16    (c) ResNet-152    (d) 2-layer CNN    (e) VGG-16

(f) ResNet-152    (g) 2-layer CNN    (h) VGG-16    (i) ResNet-152

Figure 2: Comparison of different hyperparameter optimization algorithms for 2-layer CNN, VGG-16 and ResNet-152 sharing the same legend. (a)-(c): Test error. (d)-(f): Suboptimality. (g)-(i): $\|\nabla f(\lambda)\|_2$. (Larger figures can be found in the supplement material.)

Specifically, the training of modern DNN is usually an intriguing process, involving multiple heuristic hyperparameter schedules, *e.g.* learning rate with exponential weight decay. Instead of intuitive settings, we propose to apply epoch-wise learning rates and jointly optimize these hyperparameters. The experiments are conducted on CIFAR-10 dataset with $50,000$ samples. To demonstrate the scalability of HOZOG, three deep neural networks with various structure are used, including (1) two layers DNN (2-layer CNN) with convolutional, max pooling, and normalizing layers; (2) VGG-16 (Simonyan & Zisserman, 2014), (3) ResNet-152 (He et al., 2016). The initialization of inner problem is randomized for different meta-iterations to avoid the potential dependence on the quirks of particular settings. In detail, for all experiments we apply 50 meta-iterations and optimize inner problems using stochastic gradient descent, with batch size of 256. On CNN, 100 epochs for inner problem are used, which indicates 100 hyperparameters are involved. On VGG-16, the original model takes $224 \times 224$ images as inputs, and we adjust the size of the first fully-connected layer from $7 \times 7$ convolution to $1 \times 1$ to fit CIFAR-10 inputs. Here 20 epochs for inner are used. On ResNet-152, similar processing is exploited and the inner epoch is 10.

**Results and discussions:** The results are summarized in Figure 2. The experimental results show that the learning rates computed by HOZOG achieve the lowest test error and the fastest descending speed compared to baselines on all tasks. Moreover, the proposed method requires much less time to attain the best hyperparameters, and tends to have smaller variances in gradients. It is noteworthy that, some state-of-the-art hyperparameter optimization approaches (including HOAG, REV and FOR) are missing in this setting, due to the algorithms of REV and FOR are limited to smooth functions and the implementation of HOAG is limited to the hyperparameter optimization problems with a small number of hyperparamters. However, these difficulties are avoided by our HOZOG, which also demonstrates the flexibility of HOZOG. Moreover, as a brutal search method, the performance of RS is very unstable, which can be identified from the hyper-gradients. For BO and BOHB, the instability also exists, potentially due to the highly complexity of the network structure. Another noteworthy problem with respect to BO and BOHB is the computational overhead in sampling, which make the meta-iteration extremely time consuming, compared to other methods.

We observe that the difficulty of this problem mainly comes from model complexity, instead of hyper-parameter numbers. For CNN with 100 hyper-parameters, HOZOG shows advantages in both time and suboptimality, although baselines can also efficiently find a reasonable solution. For VGG-16 and ResNet-152, we notice that though the size of hyperparameters is reduced, it takes baselines longer time to find acceptable results. Instead, HOZOG still shows fast convergence empirically. This observation indicates that HOZOG is potentially more suitable for hyperparameter optimization in large DNN.

### 3.3 DATA HYPER-CLEANING

**Experimental setup:** We evaluate HOZOG on tuning the hyperparameters of data hyper-cleaning task. Compared with the preceding problems, the data cleaning task is more challenging, since it has more hyperparameters (hundreds or even thousands).

Assuming that we have a label noise dataset, with only limited clean data provided. The data hyper-cleaning task is to allocate a hyperparameter weight $\lambda_i$ to a certain data point or a group of data points to counteract the influence of noisy samples. We split a certain data set into three subsets: $\mathcal{D}_{tr}$

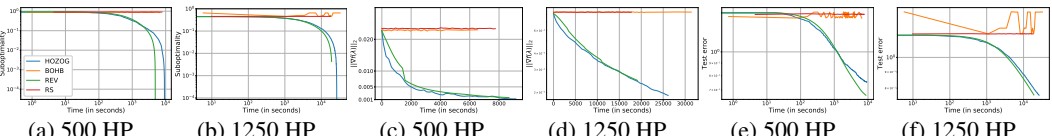

| (a) 500 HP | (b) 1250 HP | (c) 500 HP | (d) 1250 HP | (e) 500 HP | (f) 1250 HP |

Figure 3: Comparison of different hyperparameter optimization algorithms for data hyper-cleaning sharing the same legend, where "HP" is the abbreviation of hyperparameters (a)-(b): Suboptimality. (c)-(d): $\|\nabla f(\lambda)\|_2$. (e)-(f): Test error. (Larger figures can be found in the supplement material.)

of $N_{tr}$ training samples, $\mathcal{D}_{val}$ of $N_{val}$ validation samples and a test set $\mathcal{D}_t$ containing the $N_t$ samples. We set random labels to $\lceil 0.5 * N_{tr} \rceil$ training examples, and select a random subset $\mathcal{D}_f$ from $\mathcal{D}_{tr}$.

Similar to Franceschi et al. (2017), we considered a plain softmax regression model with parameters $W$ (weights) and $b$ (bias). The error of a model $(W, b)$ on an example $(x, y)$ was evaluated by using the cross-entropy $l(W, b, (x, y))$ both in the training objective function, $L$, and in the validation one, $E$. We added in $L$ an hyperparameter vector $\lambda \in \mathbb{R}^{N_h}$ that weights each group of examples in the training phase through sigmoid function, *i.e.* $L(W, b) = \frac{1}{N_{tr}} \sum_{g \in \mathcal{G}} \sum_{i \in g} \text{sigmoid}(\lambda_g) l(W, b, (x_i, y_i))$, where $\mathcal{G}$ contain $N_h$ groups random select from $\mathcal{D}_{tr}$. Thus, we have the hyperparameter optimization problem as follows.

$$\arg\min_{\lambda \in \mathbb{R}^{N_h}} E(W(\lambda), b(\lambda)), \quad s.t. \ [W(\lambda), b(\lambda)] \approx \arg\min_{W, b} L(W, b) \quad (6)$$

We instance two subset dataset for the MNIST dataset, with $N_{tr} = 5000$, $N_{val} = 5000$, $N_t = 10000$, $N_h = 1250$ and $N_{tr} = 1000$, $N_{val} = 1000$, $N_t = 4000$, $N_h = 500$. We use a standard gradient descent method for the inner problem with fixed learning rate 0.05 and 4000 iteration. RS is used as baseline method, and BOHB and REV are used as comparison.

**Results and discussions:** Figure 3 presents the results of HOZOG, BOHB, REV and RS for data hyper-cleaning. Note that the methods of GPBO, FOR and HOAG are missing here, because the hyperparameter size is beyond the capability of their implementations. The results show that HOZOG can beat RS and BOHB easily, while not perform completely as good as REV in the long run. This is because REV is an exact gradient method whose convergence rate is faster than the one of zeroth-order gradient method (*i.e.*, HOZOG) by a constant whose value is depending on $p$ (Nesterov & Spokoiny, 2017). However, computing the exact gradients in REV is costly. Specifically, REV takes about 40 seconds to finish the computation of one hyper-gradient under the setting of 1250 hyperparameters, which is only about 24 seconds for HOZOG. This is the reason why our method converges faster than REV in the early stage of training. Importantly, the application scenarios of REV are limited to smooth functions, *e.g.*, not suitable for the experimental settings of convolutional neural networks and deeper neural networks. However, our HOZOG can be utilized to a broader class of functions (*i.e.*, continuous functions).

## 3.4 DISCUSSION: IMPORTANCE OF HOZOG

The experimental results show that the black-box optimization methods have a weak performance for the high-dimensional hyperparameter optimization problems which is also verified in a large number of existing references (Brochu et al., 2010; Snoek et al., 2012), while they have the advantages of *simplicity* and *flexibility*. On the other hand, the existing gradient-based methods (Franceschi et al., 2017; 2018) need experienced researchers to provide a customized program against the optimization algorithm and sometime it would fail, while they have the advantages of *scalability* and *efficiency*. HOZOG inherits all the benefits from both approaches in that, the gradients are computed in a black-box manner, while the hyperparameter search is accomplished via gradient descent. Especially, for high-dimensional hyperparameter optimization problems which have no customized RFHO algorithm, HOZOG currently is the only choice for this kind of problems to the best of our knowledge.

## 4 CONCLUSION

*Effectiveness, efficiency, scalability, simplicity and flexibility* (*i.e.*, E2S2F) are important evaluation criteria for hyperparameter optimization methods. In this paper, we proposed a new hyperparameter optimization paradigm with zeroth-order hyper-gradients (HOZOG) which is the first method having all these benefits to the best of our knowledge. We proved the feasibility of using HOZOG to achieve hyperparameter optimization under the condition of Lipschitz continuity. The experimental results on three representative hyperparameter (the size is from 1 to 1250) optimization tasks not only verify the result in the feasibility analysis, but also demonstrate the benefits of HOZOG in terms of E2S2F, compared with the state-of-the-art hyperparameter optimization methods.

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
