# OpenReview forum: "Optimizing Large-Scale Hyperparameters via Automated Learning Algorithm"
_ICLR.cc/2021/Conference — Reject_

### Official Review · AnonReviewer4 · 2020-10-13
**A premature attempt of applying zeroth-order optimization to high-dimensional HPO**

**Rating:** 3
**Confidence:** 5

**Review:**

This work proposes to apply zeroth-order optimization to hyperparameter tuning. Zeroth-order optimization techniques use function evaluation only to approximate the gradients, thus they are applicable to black-box functions for which the gradient is difficult to compute. For hyperparameter tuning this is the case in general. Existing hypergradient-based methods are not applicable to generic tuning tasks, and have scalability issue. Existing generic methods such as Bayesian optimization have weak scalability in number of hyperparameters. I agree with these observations and the goal of having a flexible, efficient, effective, simple and scalable solution.

While the idea of using zeroth-order optimization for HPO makes sense, the paper has not reached a sufficient depth in exploring this idea. The paper proposes to use (Nesterov & Spokoiny 2017) without modification. There are several issues: (1) The paper concludes in page 5 that "Because the A-based constrained optimization problem f($\lambda$) is continuous, we
can use the zeroth-order hyper-gradient technique to optimize f($\lambda$) (Nesterov & Spokoiny, 2017).
Nesterov & Spokoiny (2017) provided the convergence guarantee of zeroth-order hyper-gradient
method when f($\lambda$) is Lipschitz continuous as defined in Definition 3." With continuity only, the algorithm can converge to local stationary points. Straightforward application of the algorithm could violate the 'effectiveness' and 'efficiency' goal. The situation is worsened in the high-dim case when the variance of the gradient approximation is large. The experiment in Sec 3.3 supports this concern, where HOZOG underperforms REV in the long run. (2) The algorithm still has several hyperparameters to tune, including q, $\lambda$ and $\mu$. From the experiments reported in the paper, each task uses different values of q, $\lambda$ and $\mu$. That raises the question about the practicability, especially, if one needs to tune these hyperparameters, whether it can still be efficient, scalable and simple. The paper has not discussed how to tune them for a given task. The procedure of tuning is required to be discussed because 'flexibility' is indeed a goal of this paper. (3) The same question can be asked about how the initial point of hyperparameters is chosen, and the paper does not discuss that.

In addition to these issues, the paper makes a strong claim "for
high-dimensional hyperparameter optimization problems which have no customized RFHO algorithm,
HOZOG currently is the only choice for this kind of problems to the best of our knowledge." in the end of Sec 3.4. For example, there are other methods designed for high-dimensional problems:

David Eriksson et al., Scalable Global Optimization via Local Bayesian Optimization, NeurIPS'19.

N. Hansen. The CMA evolution strategy: A comparing review. In Towards a New Evolutionary Computation, pages 75–102. Springer, 2006.

M. J. Powell. A view of algorithms for optimization without derivatives. Mathematics Today-Bulletin of the Institute of Mathematics and its Applications, 43(5):170–174, 2007.

Since the paper emphasizes the importance of HOZOG in the high-dim scenario, some of these stronger baselines need to used.

A final comment is that the phrase 'automated learning algorithm' in the title is not very informative, especially as the selection of hyperparameters of HOZOG is not automated.

---

> ### Author Response · Authors · 2020-11-24
> **Fair comparison issues**
>
> Thanks a lot for pointing out the related references of other methods designed for high-dimensional problems. We did not provide the comparison to these algorithms due to the following reasons.
>
> 1. [David Eriksson et al. 2019] uses trust region to improve the scalability of standard BO algorithm. Note that the same strategy of trust region can be used in our HOZOG algorithm to further improve the scalability of HOZOG. More generally, the most popular method of improving the scalability of Gaussian process based BO algorithm is low-dimensional embedding. However, same technique can also be used in our HOZOG algorithm. Thus, we only compare with standard BO algorithm for the purpose of fair comparison.
>
>
> 2. The CMA evolution strategy is also a zeroth order technique. Different to our HOZOG algorithm, it samples directions from a non-standard Gaussian distribution. We can also utilize this technique to produce more effective directions.
>
> In general, because our HOZOG is a gradient-based algorithm, HOZOG naturally can handle high-dimensional hyperparameters optimization problems which are different to different advanced BO algorithms that use the technique of low-dimensional embedding to improve the scalability of standard BO algorithms. However, if there does not exist such a low-dimensional space, this kind of improved BO optimization algorithms cannot work for high-dimensional hyperparameters optimization problems, but we can.

---

### Official Review · AnonReviewer2 · 2020-10-23
**While the paper aims to combine the benefits of gradient-based and black-box optimization, it does not compare to relevant high-dimensional BO baselines.**

**Rating:** 4
**Confidence:** 4

**Review:**

EDIT: **post rebuttal. I'd like to thank the authors for their response. As scalability to high-dimensional hyperparameter spaces is presented as a key advantage of the method, direct comparisons to high-dimensional BO techniques would be needed. The fact that using trust regions could benefit HOZOG, or that HOZOG is a better strategy compared to TurBO and REMBO, should be demonstrated empirically. I am keeping my score to 4 as the current positioning of the paper would require direct comparisons with these baselines. **


The paper introduces HOZOG, a new HPO method combining the benefits of black-box and gradient-based optimization. This is achieved by means of a finite difference approximation that computes the gradients in a black-box fashion. By doing so, the method achieves both scalability to a large number of model parameters and hyperparameters, the latter being a bottleneck in standard BO methods. A feasibility analysis and experiments against baselines show the benefits of the proposed approach.

Positive

1. **Significance.** This is the first approach trying to take a black-box approach to tackle gradient-based optimization, inheriting the benefits of the two worlds.  I expect this can lead to follow-up work in this direction.
2. **Clarity.** The paper is generally well articulated and easy to follow. I particularly appreciated the authors clearly comparing black-box and gradient based methods in Table 1 by listing desiderata.
3. **Feasibility analysis.** The work contributes a rigorous feasibility analysis studying the use of the proposed approach in HPO.

Negative

1. **Missing baselines.** If scalability to high-dimensional hyperparameter spaces is a key selling point of the method, experiments should have focused on comparing against BO methods that are designd to scale to large dimensions. For instance, the authors claim "a lot of references have pointed out that [BO] can only handle hyperparameters from a few to several dozens". It would be helpful if the authors elaborated on this point. Is it due to the poor scalability of the Gaussian process model in high dimensions? In that case, non-GP, scalable models have been proposed (e.g., SMAC) or adaptations directly targeting high-dimensional BO problems (e.g., TurBO). Code for these methods is also publicly available but these baselines are missing from the experiments. These are specific references:

(TurBO) Eriksson et al., Scalable Global Optimization via Local Bayesian Optimization. In NeurIPS 2019;

 (REMBO) Wang et al., Bayesian Optimization in High Dimensions via Random Embeddings, In IJCAI '13 + follow-ups, including Letham et al. 2020, "Re-Examining Linear Embeddings for High-Dimensional Bayesian Optimization" ;

(SMAC) Hutter et al., 2011. Sequential Model-Based Optimization for General Algorithm Configuration.  In Proceedings of the conference on Learning and Intelligent Optimization, 2011.

2. **Reproducibility.** No curves in the results include error bars. Are experiments only based on a single run? If not, clearly defined error bars should be reported. If yes, this is not enough to draw meaningful conclusions considering that the BO algortihms are highly stochastic with many sources of variability (e.g., how they are warm-started, often done through a random initial design).

3. **Structure.** The paper is visually rather crammed and the plots are too small to be readable. The authors seem to be aware of this as they provide larger versions of the figures in the appendix. The main paper should be self-contained though and this is not an acceptable way to get around the 8-page limit.

While the paper has its merits, I believe the current version falls below the acceptance bar due to the lack of competing high-dimensional BO baselines paired with reproducibility concerns (i.e., experiments have no error bars). I believe this is promising work with valuable theoretical contributions, but the experimental evaluation is currently not sufficient to justify the proposed approach. In particular, relevant baselines for high-dimensional BO should be both discussed and compared against, and all results should be based on multiple repetitions and report confidence intervals.

Minor:

1. Typos at Page 2: "salable"  ---> scalable; "hyperparmaters" --> hyperparameters
2. Related to the "structure" point above, there is no space between "results and discussion" and the rest of the text in page 8. This is also done in several places, including captions. I'd suggest making the content more crisp instead.

---

> ### Author Response · Authors · 2020-11-24
> **Fair comparison**
>
> Thanks a lot for pointing out the related references of other methods designed for high-dimensional problems. We did not provide the comparison to these algorithms due to the following reasons.
>
> 1. TurBO [David Eriksson et al. 2019] uses trust region to improve the scalability of standard BO algorithm. Note that the same strategy of trust region can be used in our HOZOG algorithm to further improve its scalability.
>
> 2. REMBO [Wang et al., 2013] and its variants use the technique of low-dimensional embedding to improve the scalability of standard BO algorithms. Same as above, the same strategy of low-dimensional embedding can be used in our HOZOG algorithm to further improve its scalability.  On the other hand, if there does not exist such a low-dimensional space, these REMB-like algorithms cannot work for high-dimensional hyperparameters optimization problems, but we can.
>
> In general, because our HOZOG is a gradient-based algorithm, HOZOG naturally can handle high-dimensional hyperparameters optimization problems which is different to standard BO algorithm only working on small number of dimensions.

---

### Official Review · AnonReviewer3 · 2020-10-28
**Interesting method, paper needs more work**

**Rating:** 4
**Confidence:** 4

**Review:**

EDIT: **post rebuttal (could not add a comment readable by authors). Thank you for your rebuttal, I have read it and the other reviews. I agree with the other reviewers that some more baselines for high-dimensional benchmarks are required, and was not convinced by your rebuttal to those requests. I also think the aspect of non-convexity and local optima touched by Reviewer 4 warrants some discussion in the paper. I maintain my score of 4. **

The paper presents a novel method for hyperparameter optimization based on zeroth order gradient estimation. The zeroth order estimation of the gradient permits the application of the method to complex architectures, provided that the hyperparameters are continuous and the objective function is Lipschitz continuous. The method appears competitive with both gradient-based HPO methods and gradient free methods on a suite of experiments with a number of hyperparameters ranging from 1 to 1250.

Strong points:
- The method is simple and is competitive with previous gradient-free HPO methods for continuous hyperparameters.
- The method appears better or competitive with previous gradient-based HPO work.
- On the technical side the paper appears solid. I see no methodological flaws in the method.

Weak points:
- There is no comparison with gradient based methods for the deep network experiments (arguably the most important benchmark). The authors state that the methods of Franceschi et al. could not be compared because they require smooth function, but checking their paper, they only mention a smooth optimization operator, and this condition appears to be met by a gradient descent optimization operator. Perhaps I am missing something here, but further justification on this point would be appreciated.
- The description of the experiments is a bit lacking in certain regards. This diminishes the reproducibility of the work. For instance, the computational budget is not specified for the experiments. How long were benchmarks run for? Was the budget specified in wallclock time? How many function evaluations did each method complete? Did you repeat each experiment and average them?
- The paper needs a lot of work on a clarity and quality standpoint. I spotted a lot of grammar mistakes and some sentences were hard to interpret.

Recommendation:

Given the current state of the paper, I would recommend rejection. The paper needs some serious copy-editing before it is up to the standards of a first tier conference such as ICLR. I should say that I like the method proposed in the paper, it would make a nice addition to the literature on HPO -- I just think it needs more polish.

Apart from paper quality, more work/details on the following points/questions would help me increase my score for this paper:

- The results are only presented using wallclock time. A general idea of the  "number of function evaluations" achieved by each method would also be informative, a graphic comparing the performance with regards to function evaluations could be added in the appendix. This is an important point because some models take a long time to run, and whatever computation is done between function evaluations becomes meaningless in comparison.
- What is the implication of removing the approximate gradient computation of HOAG with the zeroth order gradient proposed in this paper? Is it still HOAG or does it become a new method?
- Figure 2. is a bit confusing to me. Why is the x-axis "Inner epoch" for the suboptimality figures? Is this presenting the final optimization run with the best hyperparameters found?

Extra comments:

Table 1 feels rather arbitrary, what is the threshold to say that a method is "effective" or not? Simple or not? Is BOHB really that simple compared to HOAG? Complexity / optimization guarantees in terms of p and d might provide a clearer picture here (those are generally available for GP-BO and most likely for gradient based methods?) -- just an idea.

Section 2.1: Claim that the number of hyperparameters in real problems would range from hundreds to thousands. It seems to me that problems with thousands of hyperparameters are very academic and never encountered in typical real world problems. Bring forward examples or please reformulate.

Section 2.1: I have mixed feelings about the last paragraph of this section titled Enlightenment with a cross symbol. Is this a religious reference? I would argue this has no place in a scientific paper.

Proofreading is required, examples of errors:

Page 2. paragraph 2: salable -> scalable
Page 3. (such as the deep neural network) -> (such as A deep neural network)

Multiple times in the text: hyperparmeters -> hyperparAmeters

---

> ### Author Response · Authors · 2020-11-25
> **Comparison to gradient based methods for the deep network experiments**
>
> Thanks a lot for your precious time to review our paper. I give the response to the main comments as following.
>
> 1. The reason we do not compare with gradient based methods for the deep network experiments is that the code of gradient based methods cannot be directly used to solve the hyperparameters optimization problems of  deep network in our paper, due to that it is NOT a black box algorithm, it needs a customized design for a specific problem.
>
> 2.  Actually we have provided many details of our experiments in Section 3 and Table 2. We will try our best to provide more details of the experiments in the final version. In the experiments, we do not have computational budget. Actually, it is an interesting suggestion, we will try to provide experimental results within a given computational budget.
>
> 3. We will try to improve the presentation of this paper.

---

### Official Review · AnonReviewer5 · 2020-11-07
**Promising new method with some limitations in methodology and presentation**

**Rating:** 5
**Confidence:** 4

**Review:**

### Paper summary
This paper develops a novel approach to hyperparameter optimization that combines the respective advantages of black-box and gradient-based optimization methods.
The proposed method (called HOZOG) is based on zeroth-order hyper-gradients, and is empirically demonstrated on several benchmark tasks to compare favourably against state-of-the-art hyperparameter optimisation approaches in terms of simplicity, scalability, flexibility, effectiveness and efficiency.

### Pros
- overall, the paper is well and clearly written and thus easy to follow
- the manuscript addresses an important open challenge that is relevant for many practical application domains, namely hyperparameter optimization
- to tackle this challenge, the paper proposes an approach that is novel by combining previous ideas in a new, interesting and fairly principled way
- the empirical evaluation is comprehensive and convincingly demonstrates the efficacy of the proposed approach as compared to previous methods for hyperparameter optimization

### Cons
- the proposed approach seems to be fundamentally based on the assumption that the hyperparameters (and thus the induced function) are continuous; the experiments thus also seem to only involve continuous hyperparameters; this appears rather restrictive, given that many practical hyperparameter optimization problems involve important discrete variables (e.g. architectural parameters of neural networks such as the number of layers/units, or categorical parameters such as the choice of optimiser); it would be interesting to know if it would be feasible to straightforwardly extend the proposed approach to also consider such discrete variables; if not, then this seems to be a severe limitation of the method presented; note that there are several black-box/Bayesian optimization approaches (including BOHB) that can handle discrete variables, e.g. [1,2,3]
- the paper does not provide any source code, and I am not sure if I would be able to implement the method from scratch and fully reproduce the reported results with the details mentioned in the paper alone; that being said, the authors promise to make the code available upon paper acceptance
- in the first sentence in the introduction, you claim that "Modern machine learning algorithms usually involve tuning multiple hyperparameters whose size could be from one to thousands."; you proceed to mention two examples, SVMs and DNNs, with only a hand full of hyperparameters; it is thus not clear in which scenario one would have _thousands_ of hyperparameters
- in Sec. 2.1, you claim that "a lot of references have pointed out that [black-box optimization] can only handle hyperparameters from a few to several dozens"; however, you only cite one paper for this claim, and it would be beneficial to cite more of the "lot of references"; in Sec. 3.4, you again claim that "black-box optimization methods have a weak performance for the high-dimensional hyperparameter optimization problems which is also verified in a large number of existing references", proceeding to cite only two papers (which I wouldn't consider "a large number of existing references")

[1] Oh et al., 2019, "Combinatorial bayesian optimization using graph representations"
[2] Baptista et al., 2018, "Bayesian Optimization of Combinatorial Structures"
[3] Daxberger et al., 2019, "Mixed-Variable Bayesian Optimization"

### Review summary
In summary, I believe this is an interesting, enjoyable-to-read paper that tackles an important problem by proposing a promising and effective new technique. This paper will likely be of great interest to some members of the ICLR community and might become an impactful contribution for machine learning researchers and practitioners alike. That being said, I see some issues with the generality of the method as well as the presentation of the paper. As a result, I overall recommend acceptance of this manuscript, although not too enthusiastically.

### Post-rebuttal
I thank the authors for their response, addressing some of the issues/questions I had raised.
After carefully reading the other reviews (and corresponding author responses), I agree with the valid criticisms raised (some of which I had overlooked initially), which unfortunately further dampened my enthusiasm for this work.
As a result, I am slightly lowering my score.
However, I very much appreciate the author's efforts in this very promising work, and hope that they will revise their manuscript to take into account the feedback raised in the reviews.

### Minor issues
- Fig. 1: the ordering of the compared methods differs between the reported datasets
- Experiments: the name of the MNIST dataset is typically capitalised
- there are minor grammatical errors throughout the paper; I recommend that the author thoroughly proof-read the manuscript (or have it proof-read) for language issues
- the plots are barely readable without zooming in; I strongly recommend the authors to increase the font sizes to improve their readability and clarity

---

> ### Author Response · Authors · 2020-11-25
> **Thanks a lot for your support!**
>
> Thanks a lot for your precious time to review our paper. I give the response to the main comments as following.
>
> 1. Yes. The current version of the paper only considers continuous variables. This paper is the first step of large scale hyperparameter optimization with zeroth order algorithm. We plan to use some techniques like being continuous or divide and conquer to handle discrete variables. But it is beyond the scope of this paper.
>
> 2. If paper is accepted, we would like to provide the code in GitHub.
>
> 3. In our experiments of data hyper-cleaning task, there are a lot of hyperparameters because extremely each sample can correspond to a hyperparameter. We will add this example in the introduction.
>
> 4. In the final version, we would like to provide more references of black-box optimization methods which point out the weakness of the existing black-box optimization methods.

---

### Decision · Program_Chairs · 2021-01-07
**Final Decision**

**Decision:**

Reject

**Comment:**

The paper has been actively discussed in the light of the authors’ response.
Following a strong consensus across the reviewers, the paper is recommended for rejection.
Even though the paper was, overall, found quite clear, theoretically sound and tackling a relevant problem for the ICLR community, they listed several concerns that remained unclarified after the rebuttal, e.g.,

* Important baselines missing (e.g., high-dimensional BO baselines), a concern unanimously shared across the reviewers. As an example, the fact that using trust regions could benefit HOZOG should be demonstrated empirically. The same goes for the fact that HOZOG is a better strategy compared to TurBO. Such a statement warrants an empirical validation
* Further discussion about the non-convexity and local-optima concerns (raised by reviewer 4)
* Limitation to continuous hyperparameters.

This list, together with the detailed comments of the reviewers, highlight opportunities to improve the manuscript for a future resubmission.